# Comparative Transcriptome Analysis of *Bombyx mori* (Lepidoptera) Larval Hemolymph in Response to *Autographa californica* Nucleopolyhedrovirus in Differentially Resistant Strains

Xin-yi Ding [1], Xue-yang Wang [1,2], Yun-hui Kong [1], Chun-xiao Zhao [1], Sheng Qin [1,2], Xia Sun [1,2] and Mu-wang Li [1,2,*]

[1] Jiangsu Key Laboratory of Sericultural Biology and Biotechnology, School of Biotechnology, Jiangsu University of Science and Technology, Zhenjiang 212100, China; ding_xin_yi@163.com (X.-y.D.); xueyangwang@just.edu.cn (X.-y.W.); kongyunhui1@gmail.com (Y.-h.K.); www_zcx123456@163.com (C.-x.Z.); qinsheng@just.edu.cn (S.Q.); sunxia8428@163.com (X.S.)

[2] Key Laboratory of Silkworm and Mulberry Genetic Improvement, Ministry of Agriculture and Rural Affairs, Sericultural Research Institute, Chinese Academy of Agricultural Science, Zhenjiang 212100, China

* Correspondence: mwli@just.edu.cn

**Abstract:** *Bombyx mori* nucleopolyhedrovirus (BmNPV) is a kind of pathogen that causes huge economic losses to silkworm production. Although *Autographa californica* nucleopolyhedrovirus (AcMNPV) and BmNPV are both baculoviruses, the host domains of these two viruses have almost no intersection in nature. Recently, it has been found that some silkworms could be infected by recombinant AcMNPV through a puncture, which provided valuable material for studying the infection mechanism of baculovirus to silkworm. In this study, comparative transcriptomics was used to analyse the hemolymph of two differentially resistant strains following AcMNPV inoculation. There were 678 DEGs in p50 and 515 DEGs in C108 following viral infection. Among them, the upregulation and downregulation of DEGs were similar in p50; however, the upregulated DEGs were nearly twice as numerous as the downregulated DEGs in C108. The DEGs in different resistant strains differed by GO enrichment. Based on KEGG enrichment, DEGs were mainly enriched in metabolic pathways in p50 and the apoptosis pathway in C108. Moreover, 13 genes involved in metabolic pathways and 11 genes involved in the apoptosis pathway were analysed. Among the DEGs involved in apoptosis, the function of *BmTex261* in viral infection was analysed. The *BmTex261* showed the highest expression in hemolymph and a significant response to viral infection in the hemolymph of C108, indicating that it is involved in anti-AcMNPV infection. This was further validated by the significantly decreased expression of viral gene *lef3* after overexpression of *BmTex261* in BmN cells. The results provide a theoretical reference for the molecular mechanism of resistance to BmNPV in silkworms.

**Keywords:** *Bombyx mori*; AcMNPV; transcriptome analysis; apoptosis; *testis expressed genes 261*

## 1. Introduction

Silkworm, *Bombyx mori* (*B. mori*), is one of the most important insects in developing countries because of the silkworm cocoon, which is used in the medical and food industries [1–4]. However, silkworm infection by pathogens is a common problem. For example, BmNPV causes serious losses, and the underlying defence mechanism is still unclear. AcMNPV is another representative baculovirus in insects and similar to BmNPV, except in the embedded ways [5]. Normally, both of them have strict host domains and generally do not cross-infect other host domains, but studies have found that AcMNPV can infect silkworms by puncturing [6]. This study further discussed the mechanism of the

silkworm in response to baculovirus infection. The results in this study will be beneficial for clarifying the silkworm anti-BmNPV mechanism.

The baculoviruses are a family of large rod-shaped viruses that contain a circular, double-stranded genome ranging from 80 to 180 kb. Baculoviruses are endemic to invertebrates, especially insects of the order Lepidoptera [7]. The BmNPV belongs to the Baculoviridae family and has closed circular double-stranded DNA genomes; it can only infect *Bombycidae* larvae [8]. The AcMNPV is one of the pathogens of *Spodoptera frugiperda* (*S. frugiperda*). Although it infects nearly 30 lepidopteron species, it has a narrow host range [7]. Both BmNPV and AcMNPV are highly similar. There are two distinct forms in the viral replication cycle, bud virus (BV) and occlusion-derived virus (ODV), with a similar nucleocapsid structure but differences in source, morphology, protein and capsule composition, target tissue specificity, and invasion mode. The BV is mainly produced during the early replication of baculoviruses [9–11]. BmNPV genome was over 90% identical to about three-quarters of the genome of AcMNPV. The relatedness of predicted amino acid sequences of corresponding ORFs between BmNPV and AcMNPV was about 90% [12,13]; the largest difference between the two can be observed in the embedding methods. The BmNPV is a single-grain-embedded type, whereas AcMNPV is multi-grain embedded. Despite the high homology of the two genomic sequences, the host domains of the two virus types have almost no intersection in nature. For example, BmNPV can infect *B. mori* but not *S. frugiperda*, and vice versa. However, the recombinant AcMNPV can infect some silkworm strains by puncture, which is a peculiar phenomenon in the host domain of the virus [6].

Different strains of silkworms have different resistances to BmNPV in nature [14]. In recent years, researchers have studied the molecular mechanism of silkworm resistance to BmNPV from various aspects. For example, Xue et al. used the second-generation sequencing technology to analyse the differences in gene expression of Bm5 cell lines infected with BmNPV [15]. Gene enrichment analysis showed that the expression of genes related to cytoskeleton, transcription, translation, energy metabolism, ion metabolism, and the ubiquitin–proteasome pathway changed during baculovirus infection [15]. In addition, there have been some advances in proteomics [16–18]. For example, Hsp70 protein cognate, lipase-1, and chlorophyllide A-binding protein precursor were upregulated significantly after BmNPV challenge [19]. Arginine kinase was also found to be involved in the antiviral process of *B. mori* larvae against NPV infection at the protein level [20]. Other scholars have conducted a series of studies on lncRNA in silkworm resistance to BmNPV [16,17]. lncRNAs play a role in the regulation of BmNPV proliferation by Hsp90 [16]. DElncRNAs participated in the host response to BmNPV infection via interactions with their target genes and miRNAs [17]. However, the molecular mechanism of silkworm resistance to BmNPV infection has not been clearly studied.

Several studies have focused on the infection mechanism of AcMNPV in different insects. The recombinant virus of AcMNPV could be observed in midgut columnar cells of the second instar larvae of *Spodoptera exigua* at 3 h after oral inoculation and 6 h in regenerated cells [21]. The earliest expression time of the late viral gene in the columnar cells was 12 h after inoculation, which marked the replication of the virus [21]. Washburn et al. found that the early priming of GP64 reduced the oral infection of AcMNPV [22]. In addition to intracellular interactions, baculoviruses also encode gene products that function at the cellular level and thus manipulate the physiology and structure of infected animals. The gene product ecdysis hormone UDP-glycotransferase (EGT) prolongs the larval stage by inactivating the host's molting hormone, a steroid that regulates ecdysis. Viruses with this gene deleted can effectively shorten the lethal time, making them efficient pesticides [23]. The defect of AcMNPV as a Bac-to-Bac baculovirus expression system may cause cell death [24,25]. In addition, the dynamic process of nucleoactin aggregation, induced by baculovirus p78/83 protein to propagate progeny virus, has been reported [26–28]. Studies have confirmed that the nucleocapsid protein C42 of AcMNPV recruits P78/83 and ArP2/3 to mediate the aggregation of intranuclear actin [26,27,29]. By studying the molecular

mechanism of the infection of AcMNPV-infected silkworms, we can prevent cell death caused by AcMNPV, obtaining high foreign protein yields [30].

Apoptosis is a common way of fighting off viruses [31,32]. As an "evolutionary response" to host cell apoptosis, baculoviruses carry genes that encode inhibitors of apoptosis [32]. Viral proteins can also inhibit apoptosis by inactivating proteins that bind to the apoptosis pathway [31]. Apoptosis induced by insect cells infected with baculovirus first occurs in the absence of the functional *p35* gene, which was observed in AcMNPV-infected insect S21 cells [33]. The *p35* is a direct substrate inhibitor of caspases [34]. *Testis expressed gene 261* (*Tex261*) has been identified as a testis-specific protein in mice [35], but it is different in silkworms. Hideo Taniura and others found that Tex261 modulates the excitotoxic cell death induced by NMDA receptor activation [36], indicating that *BmTex261* might be involved in cell apoptosis processes.

## 2. Materials and Methods

### 2.1. Silkworm, AcMNPV, and Sample Preparation

Silkworm strain p50 (susceptible strain) and C108 (resistant strain) were maintained in the Key Laboratory of Sericulture, School of Life Sciences, Jiangsu University of Science and Technology. The larvae were fed with fresh mulberry leaves in conditions of $26 \pm 1\ °C$, $75 \pm 5\%$ relative humidity, and a 12 h day/night cycle. The temperature was decreased to $24 \pm 1\ °C$ in the last two instars, whereas the other conditions were not changed.

A budded virus of AcMNPV with an enhanced green fluorescent protein tag (BV-Egfp) is maintained in the Key Laboratory of Sericulture, School of Life Sciences, Jiangsu University of Science and Technology. The titer of BV-eGFP was determined as described in LT et al. [37].

Silkworm larvae were injected with 2.0 µL culture medium containing BV-eGFP $(1.0 \times 10^8\ \text{pfu/mL})$ per larva on the first day of the fifth instar. The control group was injected with an equal volume of culture medium. The different tissues, including midgut, hemolymph, fat body, and malpighian tubule, were collected at 36 h after injection. The samples were frozen in liquid nitrogen, immediately powdered, and stored at $-80\ °C$ until use.

### 2.2. Library Construction, Illumina Sequencing, and Read Assembly

Transcriptome analysis was performed by Beijing Novogene Technology (Beijing, China), mainly including RNA extraction, library preparation, clustering, sequencing, and assembling. A total of 1.0 µg RNA was used for library construction. The libraries were constructed applying the NEB Next® Ultra™ RNA Library Prep Kit for Illumina® (NEB, Ipswich, Massachusetts, USA). The mRNA was purified from total RNA using poly-T oligomagnetic beads. The first chain synthesis reaction buffer (5×) was cleaved using divalent cations at high temperatures. The first-strand cDNA was synthesised using random hexamer primers and M-MuLV Reverse Transcriptase (RNase H-). Subsequently, the second-strand cDNA was synthesised using DNA Polymerase I and RNase H. The remaining overhangs were converted to blunt ends by exonuclease/polymerase activities. After the 3′ ends of the DNA fragment were adenylated, NEB Next Adaptor with hairpin ring structure was linked to prepare for hybridisation. To select cDNA fragments of preferentially 250–300 bp in length, the library fragments were purified by the AMPure XP system. Then, 3 µL of USER Enzyme (US NEB) was used to the select size and connect the adapters to cDNA at 37 °C for 15 min, with 95 °C for 5 min before PCR. Phusion High-Fidelity DNA polymerase, Universal PCR primer and Index (X) primer were used for PCR; the PCR products were purified (AMPure XP system), and library quality was evaluated by the Agilent Bioanalyzer 2100 system. The raw reads were cleaned by Fastp and evaluated by the FastQC programme [38]. Hisat2 was used to map reads to reference genome from SilkDB3.0 [39–42]. Stringtie was used to assemble the novel transcript based on the mapping results [43]. FeatureCounts was used to obtain raw count of every gene. All the programmes were used with default parameters.

### 2.3. Functional Annotation and Enrichment Analysis

To annotate unigenes which were assembled by Stringtie, sequences were searched by BLASTx against the NCBI non-redundant protein (nr) database and other databases, including the Swiss-Prot protein database, the Kyoto Encyclopedia of Genes and Genomes (KEGG) and the Cluster of Orthologous Groups (COG) databases. The R package cluster-Profile was used to perform KEGG and GO enrichment [30,31]. EuKaryotic Orthologous Groups (KOG) is a eukaryote-specific version of the Clusters of Orthologous Groups (COG) tool for identifying orthologue and paralogue proteins. The KOG classification of DEGs was performed by blast against KOG database of NCBI [44,45]. All searches were performed with an E-value $< 10^{-5}$. The other genes have been fully annotated in SilkDB3.0.

### 2.4. Identification of Differentially Expressed Genes (DEGs)

The fragments per kilobase of transcript per million fragments mapped (FPKM) were calculated to represent the expression abundance of the unigenes [32]. Feature Counts were used to count the read numbers mapped to each gene, and the FPKM of each gene was calculated based on the length of the gene and the read count mapped to this gene. The FPKM, the expected number of Fragments Per Kilobase of transcript sequence per million base pairs sequenced, simultaneously considers the effects of sequencing depth and gene length for the read count and is currently the commonly used method for estimating gene expression levels. *FPKM* was calculate based on the formula and calculated in R:

$$FPKM = \frac{ExonMappedReads * 10^9}{TotalMappedReads * Exonlength} \tag{1}$$

The DESeq2 R package (https://genomebiology.biomedcentral.com/articles/10.1186/s13059-014-0550-8, accessed on November 2020) was used to analyse the differential expression between two groups; it uses a model based on the negative binomial distribution to provide statistical routines for determining differential expressions in digital gene expression data. The resulting *p*-values were adjusted by using Benjamini and Hochberg's approach to control the false discovery rate (FDR < 0.03). Genes with an adjusted *p*-value < 0.05 by DESeq2 were assigned as differentially expressed. Genes with Log2FoldFC $\geq$ 1 were selected.

### 2.5. RNA Extraction, First-Strand cDNA Synthesis, and Real-Time Quantitative PCR (RT-qPCR)

The total RNA was extracted using RNAiso Plus according to the manufacturer's instructions, purified by the isopropanol precipitation and dissolved in DEPC water. The assessing optical density (OD) absorbance ratio of 260/280 was used to determine RNA purity. The concentration of RNA was detected using a NanoDrop 2000 spectrophotometer. The integrity of RNA was checked by 1% agarose gel electrophoresis. A total of 1.0 µg of RNA was reverse-transcribed in vitro by the PrimeScript$^{TM}$ RT reagent kit according to the manufacturer's instructions.

RT-qPCR was used to detect the gene expression levels. The specific primers used in RT-qPCR were designed by the NCBI Primer-BLAST software (https://www.ncbi.nlm.nih.gov/tools/primer-blast/, accessed on January 2021) and are shown in Table 1. The reaction mixtures were prepared using the NovoStart$^{®}$SYBR qPCR SuperMix Plus kit (Novoprotein Technology Ltd., China) according to the manufacturer's instructions. Briefly, a 15 µL qPCR reaction system was used, including 7.5 µL of 2 × NovoStart$^{®}$SYBR qPCR SuperMix Plus, 0.5 µL of upstream and downstream primers, 1.5 µL of the template, and 5.0 µL of ddH$_2$O. The reactions were performed on the LightCycler$^{®}$ 96 System (Roche, Basel, Switzerland). The following qPCR protocol was used: one cycle at 95 °C for 5 min, followed by 40 cycles at 95 °C for 20 s, and 60 °C for 60 s. The $2^{-\Delta\Delta CT}$ method was adopted to calculate the relative expression level. Each group was repeated three times. *B. mori* glyceraldehyde-3-phosphate dehydrogenase (*BmGAPDH*) was used as the reference gene. The *late expression factor 3* (*lef3*) was used to detect AcMNPV (Table 1).

**Table 1.** List of primers used in this study.

| No. | Primer Names | Forward Primer (5′-3′) | Reverse Prime (5′-3′) |
|---|---|---|---|
| 1 | BMSK0012268 | CAGGCGATGAAGCTGGAGAA | GCGGACTTCCTCGTTTACCT |
| 2 | BMSK0012667 | GGCGAAGCAAAATGGCAGAA | ATTTGACGCGCTTATCGTGC |
| 3 | BMSK0013103 | CCAACTCAGCTAGACGATGCC | GATGCCAAGTTCCCGAAGATAG |
| 4 | BMSK0005642 | AACTCTGGCCGCTAAGTTCA | TCAGCTGCTCGTCCAATTCC |
| 5 | BMSK0015907 | AAAGACCAACGGAACTGCGA | CCTGTGAATTCGGTCCCCTC |
| 6 | BMSK0013101 | ACCGCACGGGAACTAGGA | CCAAGCCTAGATGCTCGTTGT |
| 7 | BMSK0000440 | GCAGTTCCGGTGAAGAGACA | AAGAAGGAGGTGGGAAGGGA |
| 8 | BMSK0008569 | AAAACACGCCCGATTCACAC | CGCGACTGTAAGTGGGAGAA |
| 9 | BMSK0009445 | TGCTACAGACGAGACTACCC | TGGATCTGTTCGCCCCTT |
| 10 | BMSK0014619 | CCGACATTGTTTGCCGTTGT | GCACTTCTGGTTGATGATGCC |
| 11 | BMSK0000576 | TAAACAAGGTCGGTCACGCA | GCCGTTTTGAACTGTGGCTT |
| 12 | *BmTex261* | CGTGTTGCCAACGACAGAAG | CGCTTTCTTGTTCCGGTGAG |
| 13 | *BmGAPDH* | CCGCGTCCCTGTTGCTAAT | CTGCCTCCTTGACCTTTTGC |
| 14 | *lef3* | CAAACGCGTTGCTTCGTACA | TGCTCGAGTCGGAAGAGGTA |
| 15 | *BmTex261 KX* | GGG<u>GGTACC</u>ATGTTATTCTTGTATTTATTGAGTTATT | GC<u>TCTAGA</u>GAACGCTTTCTTGTTCCG |

Notes: No. 1–11 are primers for RT-qPCR validation of differentially expressed transcript; No. 12–14 are primers to detect gene expression by RT-qPCR; No. 15 is the primer to amplify the CDS of *BmTex261*.

### 2.6. Construction of pIZT-mCherry-BmTex261 Overexpression Vector

The primers that amplify the coding sequence of *BmTex261* were designed with reference to NCBI-designed primers (Table 1; underlined bases are the restriction sites). The sequence was amplified from cDNA of BmN cell, and the purified products were cloned with the pMD-19T vector and sequenced by SUNYA Biotechnology (Zhejiang, China). The sequence with the construct without mutations was inserted into the pIZT/V5-His-mCherry expression vector with T4 DNA ligase (TaKaRa Biotechnology Co. Ltd., Kyoto, Japan) and confirmed by enzyme cutting via *Kpn* I and *Xba* I (TaKaRa Biotechnology Co. Ltd., Kyoto, Japan).

### 2.7. Synthesis of siRNA

To knockdown *BmTex261*, two targets, located in the functional domain, were designed by Thermo Fisher Scientific BLOCK-iT™ RNAi Designer (https://rnaidesigner.thermofisher.com/rnaiexpress/, accessed on May 2020). The siRNAs' oligos were synthesised by SUNYA Biotechnology (Zhejiang, China) and are listed in Table 2. The siRNAs were synthesised by the In Vitro Transcription T7 Kit (TaKaRa Biotechnology Co. Ltd., Japan) according to the manufacturer's instructions; siRNA, which was synthesised by BmTex261-1 Olig (siTex261), was used to knockdown the expression of *BmTex261* in BmN cells, and the siRNA which was synthesised by RFP-Olig was used as the control. The kit uses the T7 RNA polymerase, which takes the linear double-stranded DNA containing the T7 promoter sequence as the template and can transcribe the DNA sequence downstream of the promoter to efficiently synthesise single-stranded RNA. The absorbance ratio of 260/280 and the concentration were detected by a NanoDrop 2000 spectrophotometer (Thermo Scientific, New York, NY, USA). The quality of synthesised siRNAs was checked by 3% agarose gel electrophoresis at 160 V for 8 min. The qualified siRNAs were stored at −80 °C until use.

**Table 2.** List of primer sequences used to synthesise siRNA.

| Primer Names | Sequences (5′-3′) |
|---|---|
| BmTex261-1 Olig-1 | GATCACTAATACGACTCACTATAGGGAAGTCATCACGTATGCTGTATTT |
| BmTex261-1 Olig-2 | AAATACAGCATACGTGATGACTTCCCTATAGTGAGTCGTATTAGTGATC |
| BmTex261-1 Olig-3 | AAAAGTCATCACGTATGCTGTATCCCTATAGTGAGTCGTATTAGTGATC |
| BmTex261-1 Olig-4 | GATCACTAATACGACTCACTATAGGGATACAGCATACGTGATGACTTTT |
| BmTex261-2 Olig-1 | GATCACTAATACGACTCACTATAGGGAACGTTCTGACGGATTATCTGTT |
| BmTex261-2 Olig-2 | AACAGATAATCCGTCAGAACGTTCCCTATAGTGAGTCGTATTAGTGATC |
| BmTex261-2 Olig-3 | AAAACGTTCTGACGGATTATCTGCCCTATAGTGAGTCGTATTAGTGATC |
| BmTex261-2 Olig-4 | GATCACTAATACGACTCACTATAGGGCAGATAATCCGTCAGAACGTTTT |
| RFP-Olig-1 | GATCACTAATACGACTCACTATAGGGGCACCCAGACCATGAGAATTT |
| RFP-Olig-2 | AAATTCTCATGGTCTGGGTGCCCCTATAGTGAGTCGTATTAGTGATC |
| RFP-Olig-3 | AAGCACCCAGACCATGAGAATCCCTATAGTGAGTCGTATTAGTGATC |
| RFP-Olig-4 | GATCACTAATACGACTCACTATAGGGATTCTCATGGTCTGGGTGCTT |

*2.8. BmN Cell Culture and Transfection*

The BmN cell line was obtained from silkworm ovary. It was cultured in TC100 medium pH 6.2 with 10% fetal bovine serum (FBS) and 1% penicillin-streptomycin solution at 28 °C. The culture medium was replaced every 4 days.

The overexpression plasmid and dsRNA were transfected by Neofect$^{TM}$ DNA transfection reagent according to the manufacturer's instructions. The BmN cells were cultured in the dish. Briefly, each 60 mm dish required 4.0 μg plasmid or dsRNA. The plasmid and dsRNA were mixed with 200 μL TC100 without FBS and 4.0 μL Neofect transfection reagent. The highest transfection efficiency was determined for different durations.

*2.9. Statistical Analysis*

The statistical differences among three biological duplicates were determined with ANOVA and Tukey's tests, using graphpad prism. The level of statistical significance was set at *, $p < 0.05$, **, $p < 0.01$, and ***, $p < 0.001$.

**3. Results**

*3.1. Analysis of AcMNPV Infection in Differentially Resistant Strains*

In previous studies, we found that the recombinant AcMNPV-eGFP could be detected in p50 by puncture but not the C108 strain [37]. The epidermis of p50 infected with AcMNPV gradually changed from white to transparent, and the p50 could still spin silk but had abnormal cocooning and could not metamorphose into pupae (Figure 1A). All silkworms (100%) infected with AcMNPV showed this phenomenon (Figure 1A), in contrast to the p50 infected with BmNPV. The other three groups were consistent with normal silkworms (Figure 1B–D). After p50 was infected with BmNPV, the body swelled up and the epidermis ruptured, resulting in the death of p50 and the exudation of white pus. Moreover, infection with AcMNPV in silkworm individuals was also determined at different durations in a previous study; the peak of AcMNNPV proliferation occurred at 36 h after inoculation with BV-eGFP ($1.0 \times 10^8$ pfu/mL) in p50 strains [37], and this time point was selected for further sample preparation.

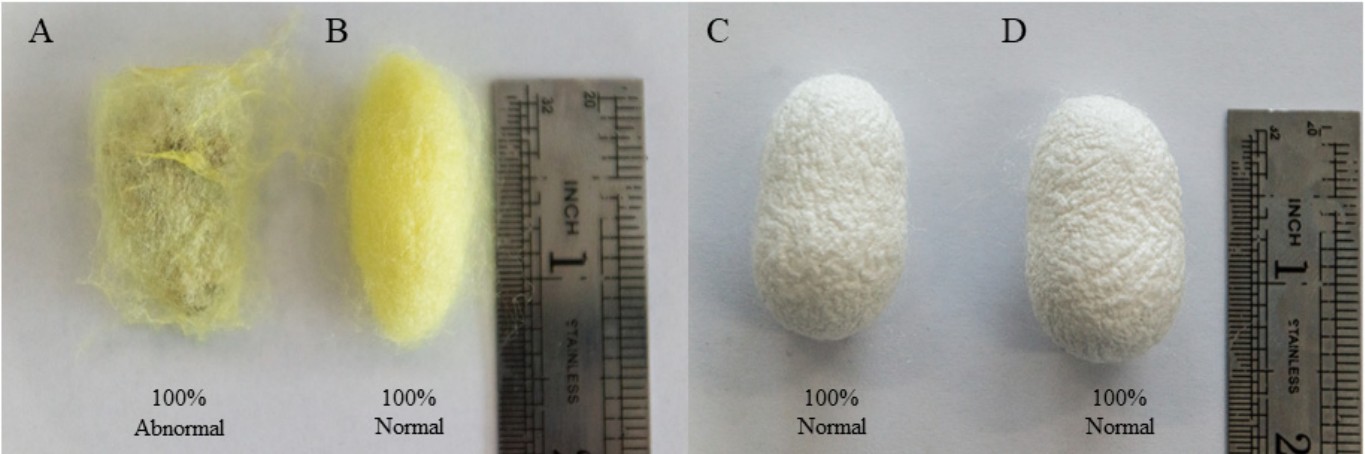

**Figure 1.** Phenotype of different resistant strains after AcMNPV inoculation. (**A**) p50 was infected with AcMNPV, (**B**) p50 without treatment, (**C**) C108 was infected with AcMNPV, and (**D**) C108 without treatment.

### 3.2. Overview of the Transcriptome of Silkworm Hemolymph

To systematically investigate the molecular mechanism of silkworm response against AcMNPV infection, transcriptome analysis was performed on the hemolymph of differentially resistant silkworm strains following AcMNPV inoculation at 36 h. Four cDNA libraries were constructed, including p50− (control group), p50+ (inoculated with AcMNPV), C108− (control group), and C108+ (inoculated with AcMNPV). The GC content of the four libraries was about 50%, and all Q30% were equal to or greater than 93.27% (Table 3). A total of 181,323,874 reads were obtained by analysing high-quality sequences (Table 3). The results indicated that the sequencing data were qualified and could be further analysed. Principal component analysis (PCA) analysis showed that the sequencing data of p50− were significantly different from the other two, and subsequent analyses excluded this group (Supplementary Figure S1). To keep the remaining two sets of data available, the data were re-analysed again by the bio-company, and the results showed the data could be used for further analysis.

**Table 3.** Summary statistics for silkworm genes based on RNA-seq data.

|  | p50− | p50+ | C108− | C108+ |
|---|---|---|---|---|
| Total Reads | 44,728,956 | 46,036,772 | 46,693,816 | 43,864,330 |
| GC Content (%) | 48 | 46 | 48 | 48 |
| % ≥ Q30 (%) | 93.35 | 93.36 | 93.27 | 94.49 |
| Mapped Reads | 41,118,461 | 33,412,593 | 39,765,815 | 42,254,696 |
| Mapped Ratio (%) | 91.92 | 72.72 | 90.66 | 90.46 |
| Unique Mapped Reads | 34,225,042 | 30,271,861 | 37,902,000 | 39,895,987 |
| Unique Mapped Ratio (%) | 85.00 | 65.87 | 86.41 | 85.44 |

Note: Q30 means the percentage of bases with mass value ≥ 30.

### 3.3. RT-qPCR Validation of Differentially Expressed Transcripts

To verify the reliability of transcriptome data, the relative expression levels of 11 randomly selected DEGs were analysed by RT-qPCR (Figure 2A–K). The trends of differential expression of these genes were consistent with the transcriptome data. Linear regression was performed for the correlation between RT-qPCR and RNA-seq data (Figure 2L); $R^2$ was 0.896, with a slope of 1.18 (Figure 2L), indicating a positive correlation between RT-qPCR and transcriptome data. In summary, we consider the data of transcriptome to be credible.

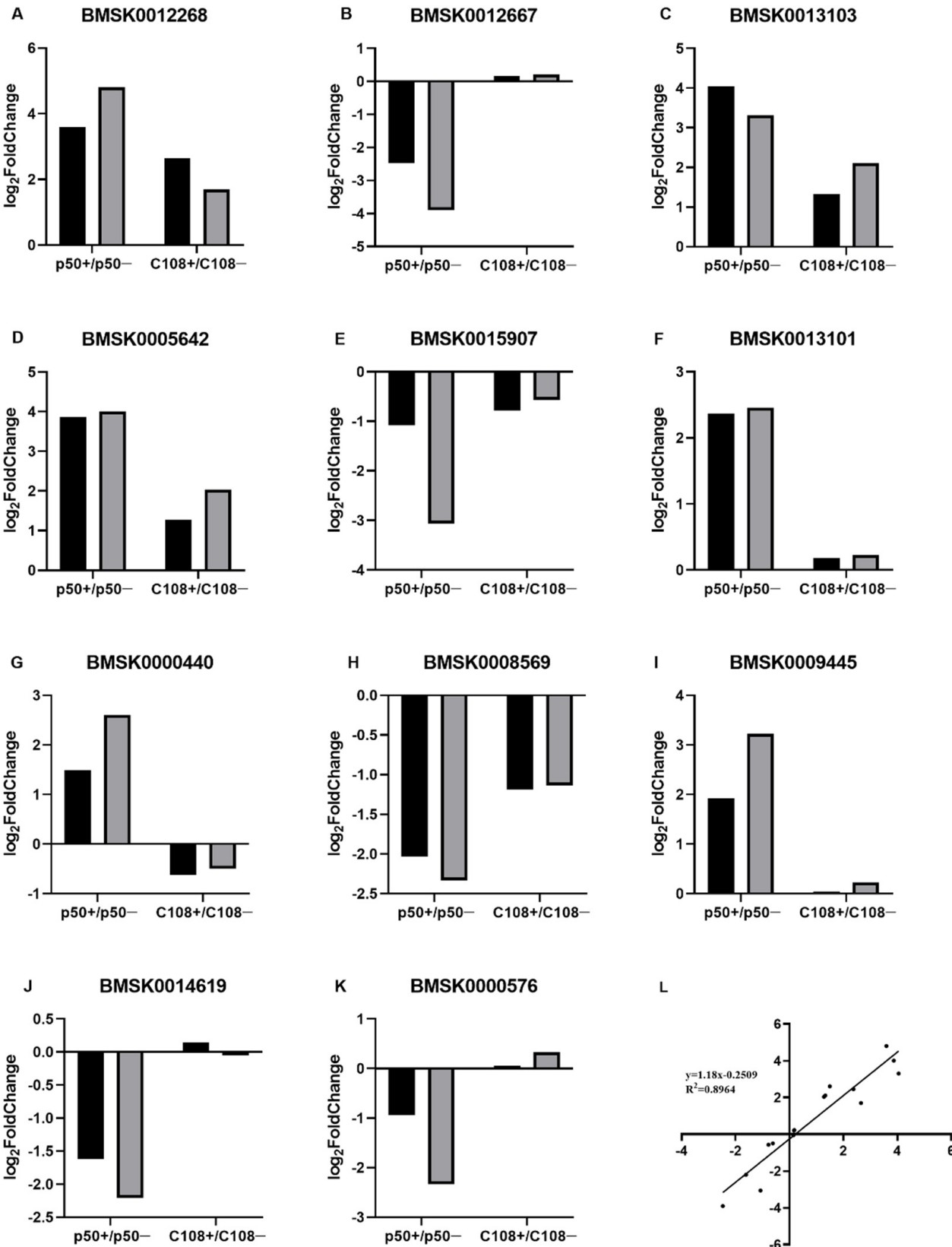

**Figure 2.** Correlation analysis between gene expression ratios obtained from transcriptome data and RT-qPCR. (**A–K**) Expression ratios (log$_2$ fold change) were obtained from RT-qPCR and transcriptome data. (**L**) Lineage analysis between the transcriptome and RT-qPCR. The ratios obtained by RT-qPCR (*x*-axis) were plotted against the ratios obtained by RNA-Seq (*y*-axis).

### 3.4. Detection and Enrichment Analysis of DEGs

The DEGs with significant differences were analysed from transcriptomic data when the difference was more than two times, and the adjusted $p$ value was less than 0.05; a total of 1402 genes were generated. In the p50+ vs. p50− group, it resulted in 691 DEGs, including 391 upregulated and 313 downregulated DEGs (Table 4). In the C108+ vs. C108− group, there were 515 DEGs, with 336 upregulated and 179 downregulated DEGs (Table 4). All DEGs have been listed in Supplementary Figures S1 and S2.

**Table 4.** Statistical analysis of DEGs in two strains following inoculation.

| Groups | DEGs | Ratio of Total Transcripts | Upregulation | Downregulation |
|---|---|---|---|---|
| p50+ vs. p50− | 679 | 6.73% | 301 (44.33%) | 378 (55.67%) |
| C108+ vs. C108− | 515 | 5.04% | 336 (65.24%) | 179 (34.76%) |

The GO enrichment analysis showed that in silkworms with different resistances, DEGs were enriched in different categories (Figure 3). In the biological process, DEGs were mainly enriched in binding activity, but there were different binding types. Some cellular components, such as vesicle, endosome, and biological regulation in molecular function, exhibited opposite expression patterns of up- and downregulation in the different resistant strains; they were upregulated in p50+ vs. p50− and downregulated in C108+ vs. C108−.

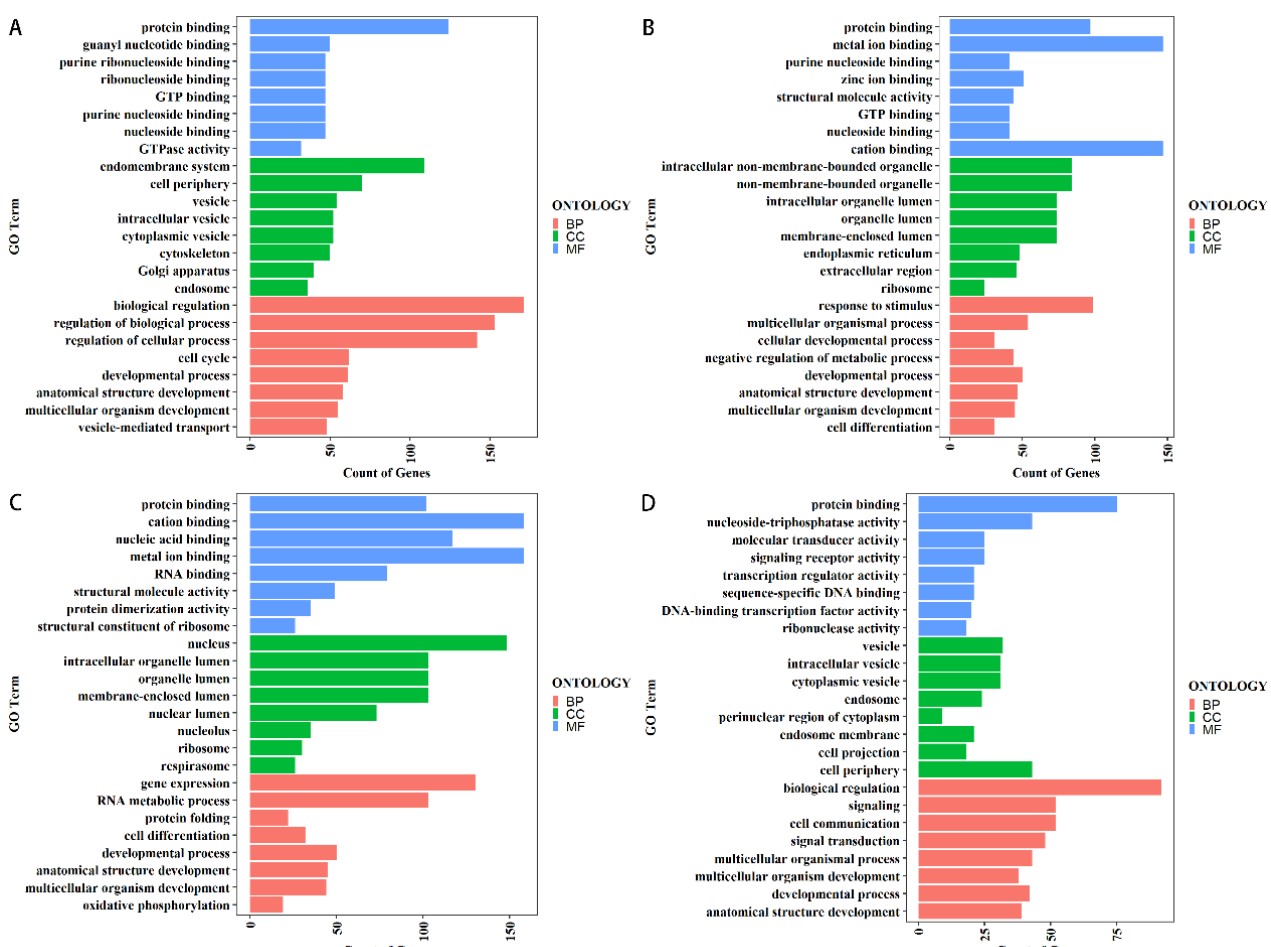

**Figure 3.** GO analysis of DEGs in differentially resistant strains following inoculation. These genes were divided into three groups based on cellular components. Biological process (BP), cellular component (CC), and molecular function (MF). (**A**) Upregulated DEGs in p50+ vs. p50− group, (**B**) upregulated DEGs in C108+ vs. C108− group, (**C**) downregulated DEGs in p50+ vs. p50− group, (**D**) downregulated DEGs in C108+ vs. C108− group.

The KEGG enrichment showed that DEGs mainly enriched in 16 metabolic pathways in p50+ vs. p50− (Figure 4A,C). For example, sulfur metabolism and linoleic acid metabolism were the most two enriched terms in p50; 10 DEGs in C108+ vs. C108− were enriched in the apoptosis pathway (Figure 4D).

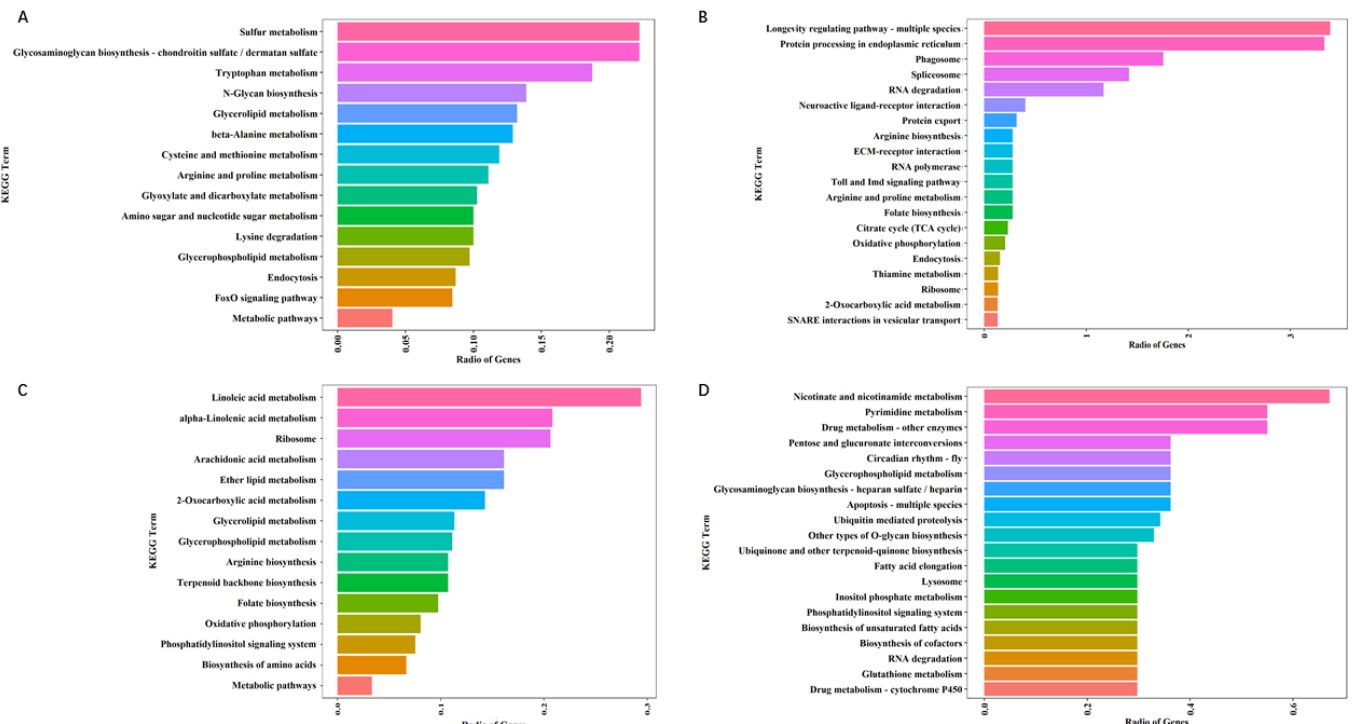

**Figure 4.** KEGG pathway enrichment analysis of DEGs in differentially resistant strains following inoculation. (**A**) Upregulated DEGs in p50+ vs. p50− group, (**B**) upregulated DEGs in C108+ vs. C108− group, (**C**) downregulated DEGs in p50+ vs. p50− group, (**D**) downregulated DEGs in C108+ vs. C108− group.

The KOG annotation showed that six metabolism-related families in p50 were more than those in C108, especially in carbohydrate transport and metabolism, and inorganic ion transport and metabolism (Figure 5). In addition, DEGs in p50 of translation, ribosomal structure, and biogenesis were significantly more than that in C108 (Figure 5). The results were consistent with those in GO and KEGG.

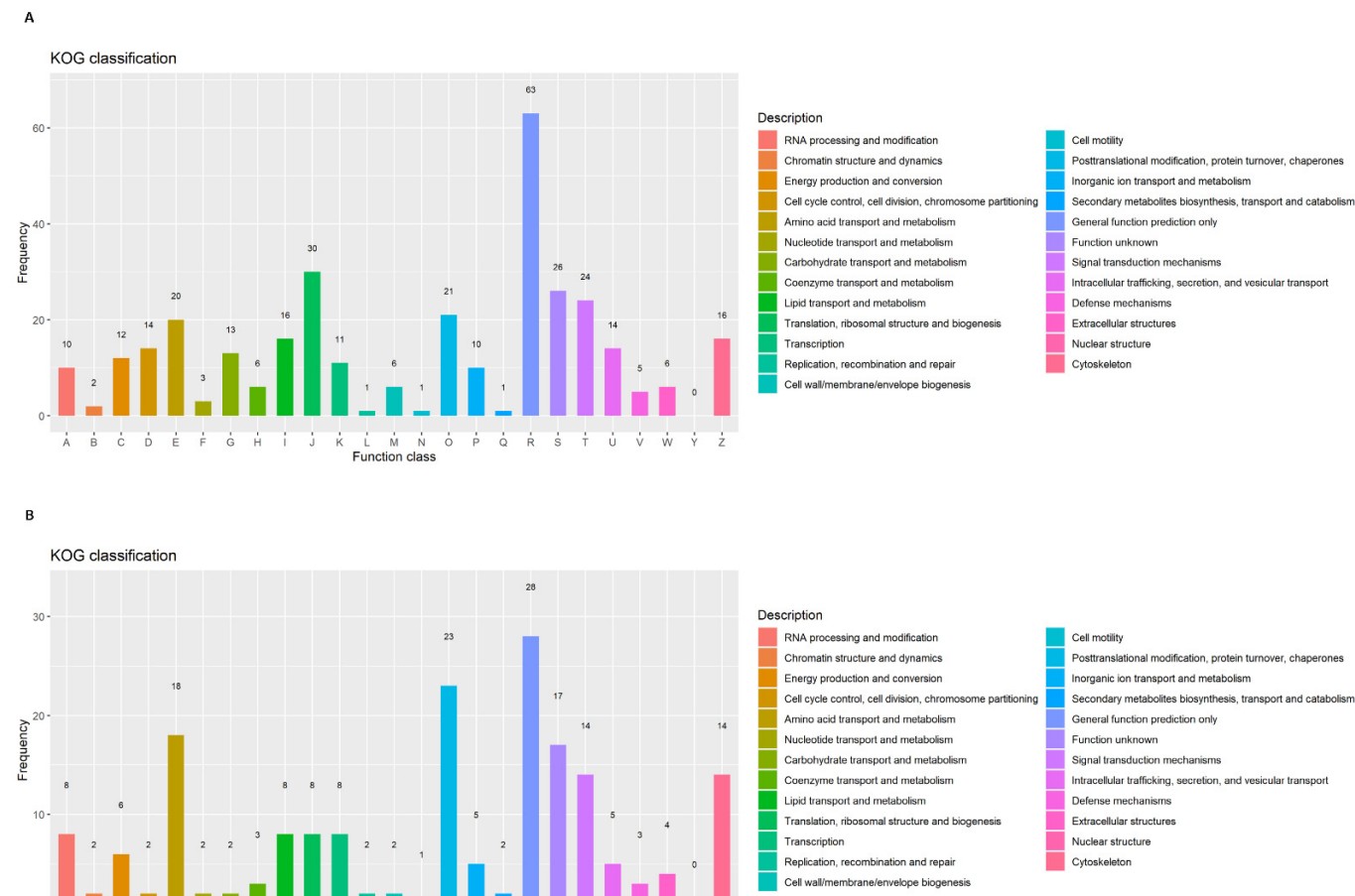

**Figure 5.** KOG annotation of DEGs in differentially resistant strains following inoculation. (**A**) DEGs in p50+ vs. p50−
group, (**B**) DEGs in C108+ vs. C108− group.

### 3.5. DEGs Involved in Metabolic and Apoptosis Pathways Showed Significant Responses to AcMNPV Infection

Based on the KEGG analysis, most DEGs in p50+ vs. p50− were enriched in the metabolic pathway. In addition, some DEGs were enriched in the apoptosis pathway. These two pathways are associated with viral infections, making it reasonable to speculate that the DEGs listed in the two pathways (Table 5) are related to the mechanism of silkworm defence to AcMNPV infection.

**Table 5.** DEGs involved in metabolic pathways and apoptosis pathways.

| Gene Name | Gene ID | p50−FRKM | p50+FPKM | C108−FRKM | C108+FRKM | p50+ vs. p50−Ratio | C108+ vs. C108−Ratio |
|---|---|---|---|---|---|---|---|
| *Metabolic pathways* | | | | | | | |
| *Catalase* | BMSK0000352 | 101.20 | 24.49 | 64.09 | 85.41 | 0.24 | 0.75 |
| *Mitochondrial aldehyde dehydrogenase* | BMSK0006974 | 41.49 | 46.17 | 152.24 | 140.61 | 1.11 | 1.08 |
| *Putative dopa decarboxylase protein* | BMSK0002058 | 3.48 | 2.290 | 14.66 | 10.90 | 0.66 | 1.34 |
| *Hydroxyacyl-coenzyme A dehydrogenase* | BMSK0002060 | 52.09 | 17.74 | 35.47 | 36.35 | 0.34 | 0.98 |
| *3-hydroxyacyl-coa dehydrogenase* | BMSK0001125 | 2.44 | 1.05 | 8.62 | 8.33 | 0.43 | 1.03 |
| *Probable 2-oxoglutarate dehydrogenase E1 component* | BMSK0003515 | 9.69 | 1.52 | 6.59 | 5.02 | 0.16 | 1.31 |
| *Kynurenine formamidase* | BMSK0008569 | 19.19 | 1.88 | 12.61 | 27.13 | 0.10 | 0.46 |
| *Mitochondrial aldehyde dehydrogenase* | BMSK0012254 | 338.85 | 59.03 | 97.36 | 121.13 | 0.17 | 0.80 |
| *Tryptophan 2,3-dioxygenase* | BMSK0008115 | 15.66 | 6.12 | 13.24 | 9.73 | 0.39 | 1.36 |
| *Inositol-trisphosphate 3-kinase A* | BMSK0005502 | 32.69 | 23.45 | 44.30 | 39.27 | 0.72 | 1.13 |
| *Phosphatidylserine decarboxylase* | BMSK0005994 | 43.74 | 10.12 | 28.99 | 32.13 | 0.23 | 0.90 |
| *UDP-glucose 6-dehydrogenase* | BMSK0014439 | 21.94 | 3.67 | 24.30 | 24.67 | 0.17 | 0.99 |
| *Glutathione S-transferase delta 3* | BMSK0003600 | 29.77 | 13.67 | 33.11 | 24.70 | 0.46 | 1.34 |
| *Apoptosis* | | | | | | | |
| *baculoviral IAP repeat-containing protein 6* | BMSK0014897 | 6.73 | 2.42 | 7.44956 | 6.170963 | 0.36 | 1.21 |
| *eukaryotic translation initiation factor 2-alpha kinase-like* | BMSK0004001 | 12.72 | 6.04 | 12.1691 | 11.05708 | 0.47 | 1.10 |
| *htra2* | BMSK0015308 | 7.03 | 2.32 | 3.154753 | 5.146924 | 0.33 | 0.61 |
| *scaffold protein salvador* | BMSK0014742 | 10.76 | 2.76 | 11.39895 | 15.83115 | 0.26 | 0.72 |
| *Ultraspiracle 2* | BMSK0001870 | 5.92 | 2.48 | 4.06216 | 8.994249 | 0.42 | 0.45 |
| *Cell division cycle protein 20 homolog* | BMSK0013179 | 10.16 | 2.35 | 6.86 | 7.00 | 0.23 | 0.98 |
| *Transcription factor E74* | BMSK0008350 | 5.72 | 1.96 | 11.45 | 14.80 | 0.34 | 0.77 |
| *Mitogen-activated protein kinase kinase kinase 7-like* | BMSK0001403 | 7.23 | 1.26 | 4.51 | 4.92 | 0.17 | 0.92 |
| *Testis expressed genes 261* | BMSK0013995 | 2.66 | 1.56 | 2.36 | 2.05 | 0.59 | 0.87 |
| *transcription factor kayak* | BMSK0014876 | 33.23 | 23.73 | 49.37783 | 56.80742 | 0.71 | 0.87 |

### 3.6. Spatiotemporal Expression Pattern of B. mori Testis Expressed Genes 261 (BmTex261)

To preliminarily study the biological function of *BmTex261*, the expression levels of *BmTex261* in different tissues of fifth instar larvae and developmental stages in p50 silkworm strain were detected by RT-qPCR. The expression level of *BmTex261* was highest in the hemolymph compared to other tissues. Moreover, *BmTex261* was relatively low among other tissues, without significant differences among tissues (Figure 6). In different developmental stages, *BmTex261* was expressed throughout all selected developmental stages, with the highest levels in the pupal stage.

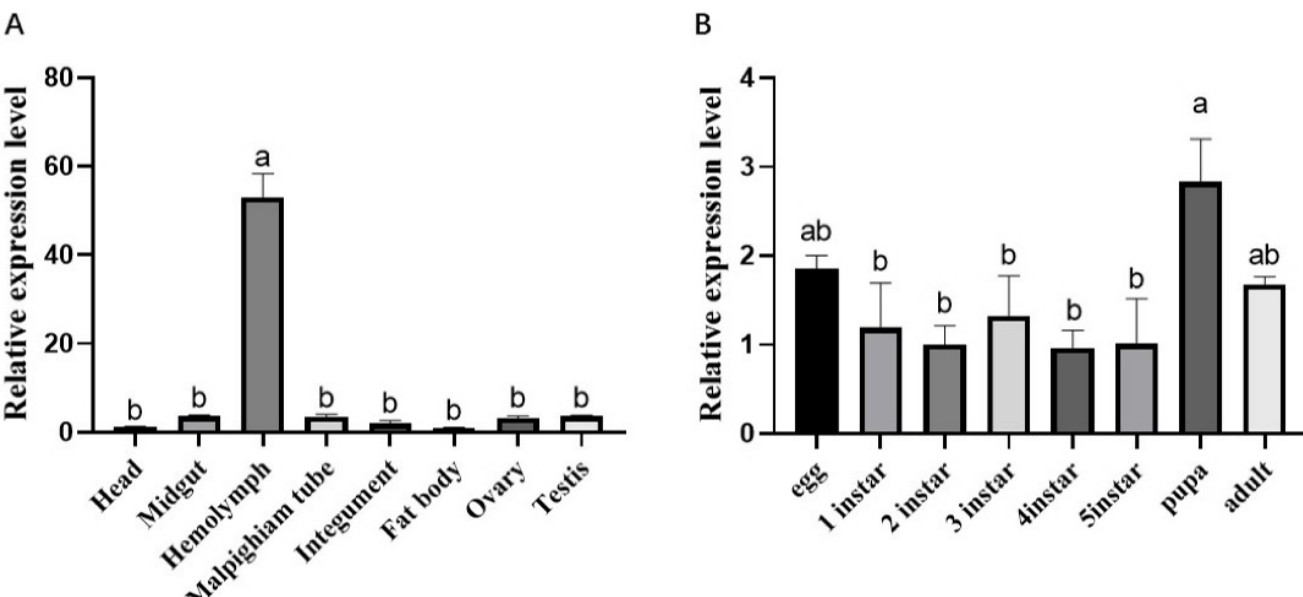

**Figure 6.** Expression analysis of *BmTex261* in different tissues and developmental stages using RT-qPCR. (**A**) Relative expression levels of *BmTex261* among different tissues; (**B**) relative expression levels of *BmTex261* among different stages. *BmGAPDH* was used to normalise the data that are shown as the mean ± standard error; the mean was obtained from three independent repeats. The $2^{-\Delta\Delta Ct}$ method was adopted to calculate the relative expression level. The statistical difference among three biological duplicates, determined with one-way ANOVA, is shown via graphpad prism. Different letters indicate statistically significant differences (a, b; $p < 0.05$).

### 3.7. Response Analysis of BmTex261 to AcMNPV Infection

The expression levels of *BmTex261* in the hemolymph, midgut, malpighian tubule, and fat body in p50 (susceptible strain) and C108 (resistant strain) were detected after injection with AcMNPV. The expression levels of *BmTex261* in the hemolymph of the p50 strain were significantly downregulated after injection with AcMNPV as compared to the control group (Figure 7), whereas the expression of *BmTex261* was upregulated in the midgut of p50 (Figure 7).

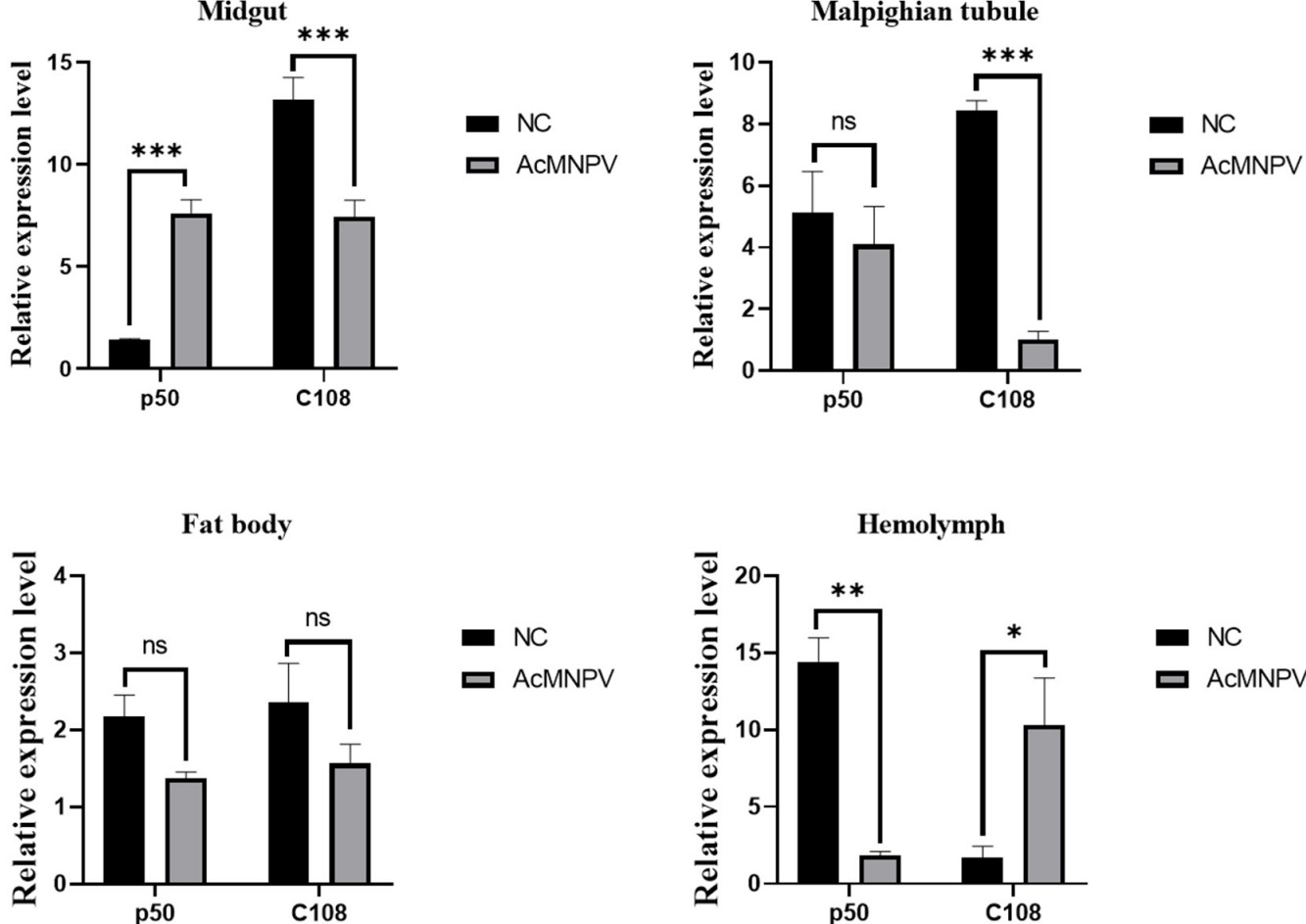

**Figure 7.** Analysis of *BmTex261* expression levels in different tissues among different resistant strains following AcMNPV infection using RT-qPCR. NC is the negative control. *BmGAPDH* was used to normalise the data that are shown as the mean ± standard error; the mean was obtained from three independent repeats. The $2^{-\Delta\Delta Ct}$ method was adopted to calculate the relative expression level. The statistical difference among three biological duplicates, determined with one-way ANOVA, is shown via graphpad prism. The level of statistical significance was set at ns, $p > 0.05$, *, $p < 0.05$; **, $p < 0.01$; and ***, $p < 0.001$.

### 3.8. Overexpression of BmTex261 Inhibited AcMNPV Infection in BmN Cells

To further explore the function of *BmTex261* in response to AcMNPV, the recombinant plasmid of pIZT-mCherry-BmTex261 was constructed (Figure 8A) and transfected into the BmN cells to overexpress *BmTex261* (Figure 8B). The stable BmN cell line contained pIZT-mCherry-BmTex261 were generated by a final concentration of 200 μg/mL zeocin. The overexpression of *BmTex261* was detected by increasing about 56-fold over the control group. Furthermore, the expression of *lef3* of AcMNPV in the overexpression group was significantly decreased as compared to the control group at 24 h, 48 h, and 72 h after AcMNPV infection (Figure 8C).

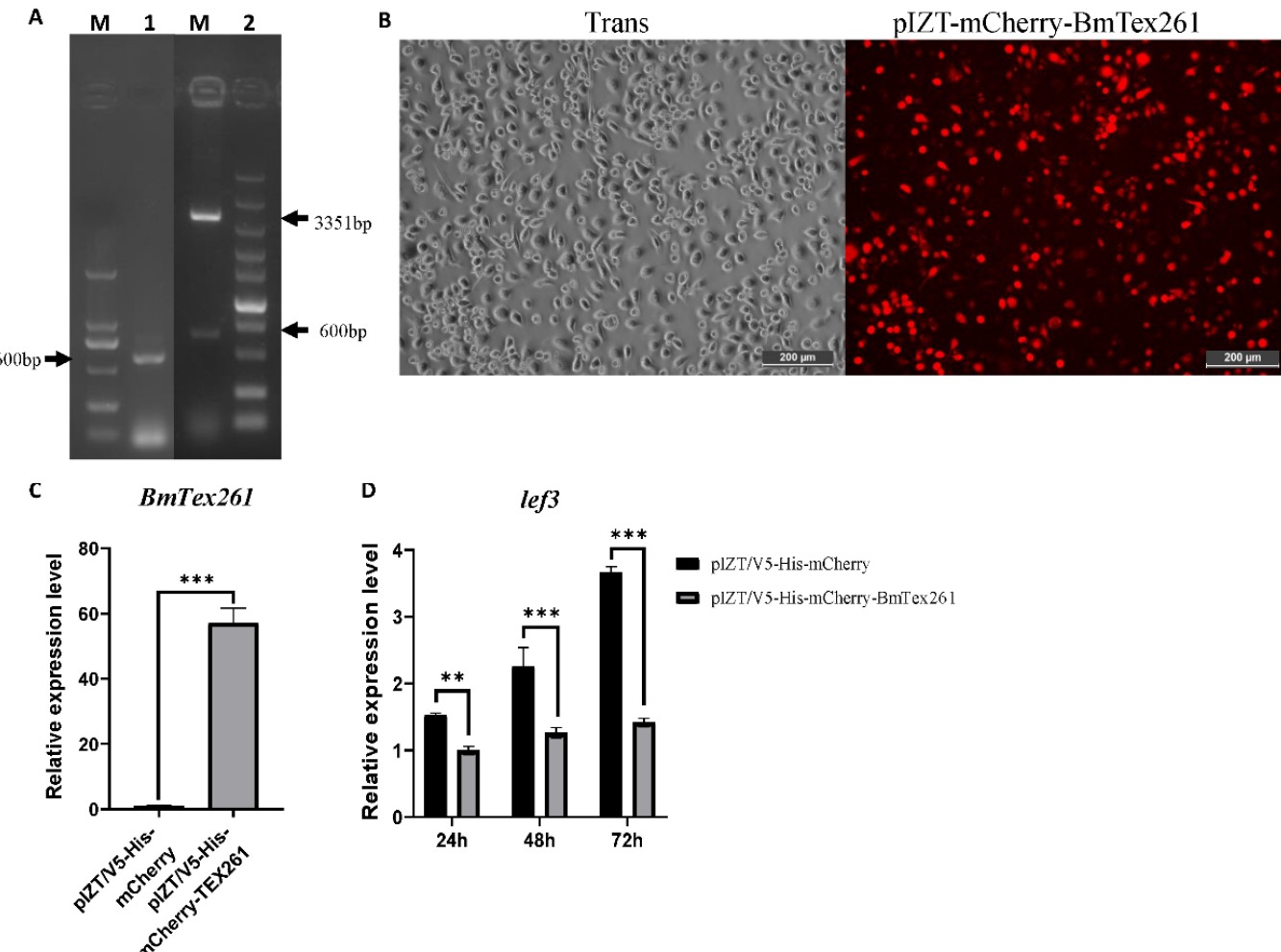

**Figure 8.** Analysis of AcMNPV infection after overexpression of *BmTex261* at different durations in BmN cells. (**A**) Construction of pIZT-mCherry-BmTex261. (1) Amplification of CDS sequence of *BmTex261* using PCR from BmN cells, the length of *BmTex261* = 600 bp; (2) validation of the recombinant vector using *Kpn* I and *Xba* I, the length of *BmTex261* = 600 bp, the length of the vector = 3351 bp; (**B**) overexpression detection of *BmTex261* after transfection with pIZT-mCherry-BmTex261 in BmN cells. Red, mCherry; scale bar = 200 μm; (**C**) expression level analysis of *BmTex261* in the stable cell line containing pIZT-mCherry-BmTex261; (**D**) analysis of *lef3* expression after overexpression of *BmTex261* at different durations. *BmGAPDH* was used to normalise the data that are shown as the mean ± standard error; the mean was obtained from three independent repeats. The $2^{-\Delta\Delta Ct}$ method was adopted to calculate the relative expression level. The statistical difference among three biological duplicates, determined with one-way ANOVA, is shown via graphpad prism. The level of statistical significance was set at **, $p <0.01$; and ***, $p <0.001$.

### 3.9. Knockdown BmTex261 Has No Effect on AcMNPV Infection in BmN Cells

To knockdown *BmTex261* at the BmN cell level, siRNAs were synthesised and transfected into BmN cells. Twenty-four hours was selected as the time point for analysis due to the best transfection efficiency, and AcMNPV was added at 24 h after transfection. The expression of *BmTex261* was successfully knocked down at different durations after transfection with siRNAs in BmN cells (Figure 9A). To further verify the relationship between *BmTex261* and AcMNPV, the expression level of *lef3* at 24, 48, and 72 h after AcMNPV infection in the BmN cells was detected, and the results showed that knockdown of *BmTex261* had no effect on AcMNPV infection (Figure 9B).

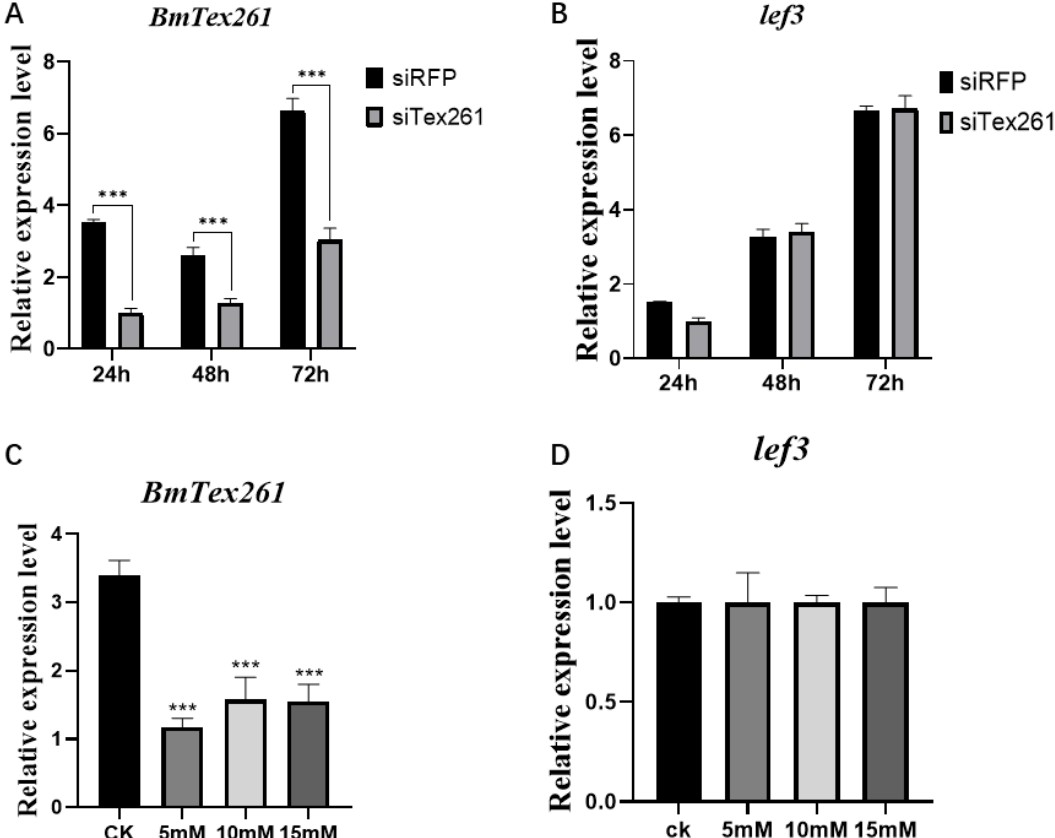

**Figure 9.** Analysis of AcMNPV infection after knocking down *BmTex261* in BmN cells. (**A**) Expression analysis of *BmTex261* after siTex261 transfection at different durations by using RT-qPCR. (**B**) Analysis of *lef3* expression after knockdown of *BmTex261* at different durations after AcMNPV infection. (**C**) Relative expression levels of *BmTex261* after adding different concentrations of CaCl$_2$; (**D**) relative expression levels of *lef3* after adding different concentrations of CaCl$_2$. *BmGAPDH* was used to normalise the data that are shown as the mean ± standard error; the mean was obtained from three independent repeats. The $2^{-\Delta\Delta Ct}$ method was adopted to calculate the relative expression level. The statistical difference among three biological duplicates, determined with one-way ANOVA, is shown via graphpad prism. The level of statistical significance was set at ***, $p < 0.001$.

Based on literature analysis, we found that *Tex261* is related to Ca$^{2+}$ [28,36,46]. In this study, the different concentrations of Ca$^{2+}$ were added to the medium; the final concentrations of 0, 5, 10, and 15 mM were used to analyse the function of *BmTex261*. The results showed that the expression of *BmTex261* was significantly downregulated after adding different concentrations of Ca$^{2+}$ (Figure 9C). Moreover, the expression of *lef3* was detected at 24 h after adding different concentrations of CaCl$_2$. The expression level of *lef3* was significantly different in this case (Figure 9D); this was consistent with the result after knockdown of *BmTex261*.

## 4. Discussion

BmNPV causes serious losses in silkworm production, and the underlying defence mechanism is still unclear. The AcMNPV is another representative baculovirus in insects and is largely similar to BmNPV. Normally, both viruses have strict host domains and generally do not cross-infect other host domains, but studies have found that recombinant AcMNPV can infect silkworms by punctures. This provides new incentives for the study of the mechanism of insect responses to baculovirus infection. In this study, transcriptome sequencing of AcMNPV infection-related genes in the hemolymph of *B. mori* was used to analyse the mechanism of silkworm defence against AcMNPV infection, which will be beneficial for clarifying the mechanism of silkworm defence against BmNPV.

In this study, there are two strains with different resistances. The susceptible strain p50 cannot pupate and dies, whereas the resistant strain showed no effect after being infected with recombinant AcMNPV on the first day of the fifth instar (Figure 1). Here, the GC content of the four libraries was about 50%, and all Q30% were equal to or greater than 93.27% (Table 3), indicating that the transcriptome data are of good quality. Correlation analysis between gene expression ratios obtained from transcriptome data and RT-PCR shows that the expression quantity is credible. Thus, the transcriptome results can be used for further studies.

Analysis of transcriptome data revealed 678 DEGs in the p50+ vs. p50− group and 515 DEGs in the C108+ vs. C108− group (Table 4). All the DEGs were listed in Supplementary Tables S1 and S2. Among them, the upregulation and downregulation of DEGs were almost similar in p50+ vs. p50− (Table 4); however, only half as many downregulated DEGs were upregulated DEGs in C108+ vs. C108− (Table 4), suggesting that AcMNPV has different effects on the two strains. Based on GO enrichment analysis of DEGs (Figure 3), we found that in the biological process, DEGs were mainly enriched in binding activity, but there were different binding types. Some cellular components, such as vesicle, endosome, and biological regulation in molecular function, differed among the different resistant strains; they were upregulated in p50+ vs. p50− and downregulated in C108+ vs. C108 (Figure 3). These results indicate that AcMNPV causes different responses in silkworms with different resistances. In addition, the trend of genes related to biological regulation in the two groups was opposite, confirming the phenomenon that AcMNPV infection of p50 inhibited pupation (Figure 1). The KEGG analysis showed that DEGs were mainly enriched in metabolic pathways; some DEGs were enriched in the apoptosis pathway (Table 5). The KOG annotation showed that the metabolism-related families in p50 were more than those in C108 (Figure 5). In addition, DEGs in p50 of translation, ribosomal structure, and biogenesis were significantly more than that in C108 (Figure 5). The results were consistent with those in GO and KEGG. Some antioxidant-related proteins were found in the tryptophan metabolic pathway, such as catalase, which acts as a barrier against viral infections [47]. In *Aedes aegypti*, viral resistance is closely related to catalase, which is upregulated after hemolymph intake to protect the body from viruses in the hemolymph [48]. Catalase was also studied in BmNPV, but it cannot stop the infection from occurring since a coevolutionary bond is maintained between the virus and the host in *B. mori* [47]. Therefore, catalase should be a related gene that affects the resistance of silkworms. In addition, glutathione S-transferases (GSTs) are believed to play a role in the detoxification of xenobiotics, the resistance to insect viruses and pesticides, intracellular transport, the biosynthesis of hormones, and the protection against oxidative stress [49].

Apoptosis has long been regarded as a defence mechanism against viral infection (Table 5). Many genes related to regulating cancer cells can regulate apoptosis in silkworms [50–53]. In the apoptotic pathway, ultraspiracle (USP) was also found, along with its downstream gene transcription factor (E74). Since they have a certain relationship with the growth and metamorphosis of silkworms [54,55], they might be related to the infection with AcMNPV. In previous studies, *Tex261* has been reported to induce cytotoxic death and was found in the apoptosis pathway [36]. Moreover, among the identified apoptosis genes, there was no report about the relationship of *Testis expressed gene 261* (*Tex261*) and virus; it is valuable to detect the role of *BmTex261* in AcMNPV infection. The expression levels of *BmTex261* were measured in different tissues and developmental stages of the p50 strain by RT-qPCR; expression of *BmTex261* was highest in the hemolymph (Figure 6A), and its immune stress response was most typical in the hemolymph after AcMNPV infection (Figure 7). These results may explain why AcMNPV can only infect some silkworms via the hemolymph by a puncture. Moreover, *BmTex261* had different expression levels following AcMNPV infection in p50, but no significant differences in the C108 midgut, fat body, and malpighian tubule were observed (Figure 7). This gene was speculated to be related to the infection of silkworms with AcMNPV. The upregulated expression levels of *BmTex261* in the midgut of C108 (Figure 7) may be a sign that AcMNPV is suppressed in the

hemolymph, and thus, the virus cannot invade other tissues. In addition, the expression level of *BmTex261* was highest in the pupal stage (Figure 6B), indicating that this gene plays an important role in the metamorphosis of silkworms and that p50 could not pupate after AcMNPV infection. To further investigate the role of *BmTex261* in AcMNPV, the gene was overexpressed in BmN cells using the pIZT overexpression vector (Figure 8A,B). Based on the results, *BmTex261* was successfully overexpressed at 36 h (Figure 8C). At 24, 48, and 72 h after the addition of AcMNPV, the expression level of the virus was significantly inhibited (Figure 8D), which leads us to infer that AcMNPV may regulate the expression level of the gene after invading silkworm cells, which reduces the resistance of silkworms and makes it impossible for the body to successfully immunise against the virus. The *BmTex261* was knocked down by siRNA in the BmN cell (Figure 9A), but the reduced expression did not affect the expression of AcMNPV (Figure 9B). This may be due to the compensatory effects of the organism. To further verify this result, we analysed the differentially expressed genes, *Tex261* and *Inositol 1,4,5-trisphosphate* (*Ins-1,4,5-P3*), which might have a certain relationship with $Ca^{2+}$ [36,56], and we therefore speculated that the calcium ion level in silkworm also had a certain change. Besides, $Ca^{2+}$ can significantly knockdown *BmTex261* (Figure 9C), and its effect on the virus is consistent with that of RNAi, suggesting that knockdown *BmTex261* can induce compensatory effects in *B. mori*.

## 5. Conclusions

In this study, DEGs were generated from transcriptomes of differentially resistant silkworm strains following AcMNPV infection, and the result of KEGG showed that DEGs in p50 were concentrated in the metabolic pathway and apoptosis pathway. Moreover, *BmTex261* belonging to the apoptosis pathway was further analysed by overexpression and RNAi, and we found that the overexpression of *BmTex261* could knockdown AcMNPV infection in BmN cells. The results in this study provide the theoretical reference for clarifying the mechanism of the infection and replication process of baculovirus in silkworms.

**Supplementary Materials:** The following are available online at https://www.mdpi.com/article/10.3390/pr9081401/s1, Figure S1: principal component analysis of the transcriptome, Table S1: DEGs in p50 strain, Table S2: DEGs in C108 strain.

**Author Contributions:** Conceived and designed the experiments: X.-y.D. and X.-y.W. Performed the experiments: X.-y.D. and C.-x.Z. Analysed the data: X.-y.D., Y.-h.K., S.Q. Contributed reagents/materials/analysis tools: M.-w.L. and X.S. Wrote the paper: X.-y.D. and X.-y.W. All authors have read and agreed to the published version of the manuscript.

**Funding:** This work was supported by the National Natural Science Foundation of China, 31772523, Postgraduate Research & Practice Innovation Program of Jiangsu Province (KYCX21_3513).

**Institutional Review Board Statement:** Not applicable.

**Informed Consent Statement:** Not applicable.

**Data Availability Statement:** The data that support the findings of this study are openly available in the SRA database at https://www.ncbi.nlm.nih.gov/sra, accessed on 2 July 2021, reference number SRR15247045-SRR15247052. The data that support the findings of this study are available in the Supplementary Materials of this article.

**Conflicts of Interest:** The authors declare no conflict of interest.

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
