# Peer review of "Comparative Transcriptome Analysis of Bombyx mori (Lepidoptera) Larval Hemolymph in Response to Autographa californica Nucleopolyhedrovirus in Differentially Resistant Strains"

_processes, doi:10.3390/pr9081401_

Round 1

Reviewer 1 Report

  1. Overall statement:

   This study investigated the transcriptomes of the hemolymph of silkworms with and without infection with BmNPV and AcMNPV viruses in 2 differentially resistant cell lines. The study looked into the DEGs and compared them, which genes were upregulated and which genes were downregulated in each cell line and condition, and annotated the genes, they were mainly found to be metabolism or apoptosis pathway genes. Then a selected gene BmTex261 was knocked down by SiRNA and different Calcium ion concentrations were used to determine if its expression is affected by Calcium ion concentrations. Overall it is a useful study that provides more information about BmNPV and AcMNPV viral infections to the silkworm hemolymph and is a good starting point for further studies.

  1. Overall strengths of the article:

   The article provides insights into the viral infection of 2 differentially resistant strains of silkworm cell lines, as to their transcriptome and role of BmTex261 gene and can be used as a basis later on to further study the moleculkar mechanism of viral infection of BmNPV and AcMNPV in silkworm cells.

  1. Specific comments on weaknesses of the article and what could be done to improve it:
    1. Major points:

  1. The transcriptomic data should be accessible, where is it deposited publicly?

  1. The sample size is actually limited, they are 3 biological replicates from each condition, and one of the controls was excluded later because it was different, will you still make its transcriptomic data available? Clarification is needed because for the condition with only 2 samples, the statistics would not be correct.

  1. The word “strains” to be “cell lines”, as it is referring to the silkworm cells, and strains is used rather for bacteria and viruses.

  1. Page 1: lines 33-35: first sentence is too vague and has only 1 reference added, so more references and specific examples would be best here.

  1. Page 2 line 41: “...new incentives...”, this sentence is not clear, the wording needs to be changed, and the following sentence could be changed to: “will further decipher the silkworm…”.

  1. Line 56: is this talking about the similarity of all ORFs in BmNPV and AcMNPV viruses? The wording is unclear and needs to be modified.

  1. Line 70 statement is too general, if the authors will keep it, then some elaboration is required, maybe they can mention one significant example.

  1. Line 82: “organic level” is unclear and needs modification, is “cellular level” meant?

  1. Between sections 2.2 and 2.3, more elaboration is needed about the bioinformatics analysis, so were the reads annotated? Or were the reads assembled? Were the reads mapped to a reference genome?

  1. Line 191: “sequence without the mutant” to be “wildtype” or “has the construct without mutations”.

  1. Line 226, please clarify what is meant by “previous studies”, does it mean the earlier experiments in this study? Re-wording is required.

  1. Re-wording of “screened” it could be “analyzed” in lines 267, 269. Also “was generated” could be “resulted into …DEGs”. Line 271: “ones” to be “DEGs”. Line 268: “P adjusted” to be “adjusted p value”. Line 267: to be “transcriptomic data”.

  1. Line 276: “were totally different” is too general, re-wording is required e.g. “exhibited opposite expression patterns of up- and down-regulation”.

  1. Table 4, the DEGs represent how much of the total transcripts detected? It is best to be mentioned. Also, was a protein family analysis done? As it could be more accurate to determine the protein function and could provide more information to the DEGs’ functions.

  1. Line 295, it is best to mention which metabolic pathways, and mention the number of the DEGs, rather than “some DEGs”, to be more accurate.

  1. Clarify why BmTex261 was selected for knockdown.

  1. Line 338 requires re-wording, “screened” seems to be not the most accurate word.

  1. Line 370: re-wording of inhibited” to “knocked down”.

  1. Gel A in figure 7 seems to come from 2 different gels, so better separate them, or please clarify. Also each length (2771 and 510 bp), what does each represent? Mention in the legend.

  1. Line 365 “different concentrations”, is it increasing concentrations”, or mention the concentrations exactly, as it is a somewhat vague statement.

  1. “Different times” to be corrected throughout the manuscript e,g. line 371, to e.g. “durations”, or another accurate word. And clarification as to what this time refers to, e.g. 24 hrs after transfection?!

  1. Line 402, please refer to the figure it is describing.

  1. Line 427, “blood”, do you mean “hemolymph”, consistent terminology is best to be used.

  1. Lines 447, 448, 449 need re-writing as they are too general and “screened out” is not clear.

  1. Finally, include a statement at the end of the discussion to state the main conclusions of this study and highlight the importance of them and what are the future prospects of it.

  1. Minor points:

  1. Grammar and typos require proofreading throughout the whole manuscript. Abstract line 6: “transcriptome” to be “transcriptomics”, “differently” to be “differentially” here and throughout the text -also in line 276-, line 12: “screened” to be “analyzed” or a similar word, as screened is not the accurate word, line 13: “virus infection” to “viral infection” throughout the text. “Series loses” to be “serious losses” throughout the text.

  1. References missing in the following lines and need to be added: Page 1, lines 35 and line 39 in the intro. Lines 65, 67. Lines 88, 93, 96. Line 227. Line 362.

  1. Page 1 line 40: “cross-infect”, seems to be an incomplete sentence, change to be “cross-infect other host domains”.

  1. Typo in line 46: to be “The BmNPV”.

  1. Table 1: title needs more description, as to which experiment these primers belong to, and clarification which is the 5’ and which is the 3’ end of the sequences. Best to add also which region is each amplifying. The last primer has small letters which is inconsistent with the rest of the table.

  1. Typo in line 195 to be “functional domain”.

  1. Typo in line 295 to be “KEGG”.

  1. Table 2 title has 2 lines, line 2 seems to be requiring to be deleted. Please revise.

Author Response

Authors' response: Thank you for your valuable comments. We absolutely agree with your viewpoints. Our point-by-point responses to your comments are presented below.

  1. The transcriptomic data should be accessible, where is it deposited publicly?

Authors' response: Thank you for your valuable comments. The raw data has already been uploaded to the SRA database (SRR15247045-SRR15247052), and the reviewer link was as here: https://dataview.ncbi.nlm.nih.gov/object/PRJNA749712?reviewer=e9ifu3264jbhqgnpu16qfn3jhh. This section has been added in lines 503-506 (page 17).

  1. The sample size is actually limited, they are 3 biological replicates from each condition, and one of the controls was excluded later because it was different, will you still make its transcriptomic data available? Clarification is needed because for the condition with only 2 samples, the statistics would not be correct.

 Authors' response: Thank you for your valuable comments. To evaluate the available of the sequencing data, the principal component analysis (PCA) was used to analyze. The result showed that one of the replications of p50- was not good enough, and it was removed. To keep the available of the remaining two sets of data, the data was re-analyzed again by the bio-company, and the results showed the data could be used for further analysis. The description was added in the line 268-270 (page 7).

  1. The word “strains” to be “cell lines”, as it is referring to the silkworm cells, and strains is used rather for bacteria and viruses.

Authors' response: We have reviewed a lot of literatures to find the description of the line of Bombyx mori, and most of them refer to "strain" as the species of silkworm. The “strain” is mean to the silkworm variety in this manuscript, too.

  1. Page 1: lines 33-35: first sentence is too vague and has only 1 reference added, so more references and specific examples would be best here.

Authors' response: Thank you for your valuable comments. We have added the references in lines 36 (page 1).

  1. Page 2 line 41: “...new incentives...”, this sentence is not clear, the wording needs to be changed, and the following sentence could be changed to: “will further decipher the silkworm…”.

Authors' response: Thank you for pointing this out. This section has been revised in lines 44-46 (page 2).

  1. Line 56: is this talking about the similarity of all ORFs in BmNPV and AcMNPV viruses? The wording is unclear and needs to be modified.

Authors' response: Thank you for your valuable comments. We are sorry for the unclear description. We have carefully read the reference you mentioned above and gained some content from it. BmNPV genome was over 90 % identical to about three-quarters of the genome of AcMNPV. The relatedness of predicted amino acid sequences of corresponding ORFs between BmNPV and AcMNPV was about 90 % [1]. The description has been added (lines 54-57, page 2).

  1. Ayres, M.D., et al., The complete DNA sequence of Autographa californica nuclear polyhedrosis virus. Virology, 1994. 202(2): p. 586-605.

  1. Line 70 statement is too general, if the authors will keep it, then some elaboration is required, maybe they can mention one significant example.

 Authors' response: This section has been revised (lines 71-79, page 2).

  1. Line 82: “organic level” is unclear and needs modification, is “cellular level” meant?

 Authors' response: We are very sorry that this was not explained clearly in the manuscript. The “organic level” is meant the “cellular level”, and this word has been revised (line 87, page 2).

  1. Between sections 2.2 and 2.3, more elaboration is needed about the bioinformatics analysis, so were the reads annotated? Or were the reads assembled? Were the reads mapped to a reference genome?

 Authors' response: Thank you for your valuable comments. We are very sorry for the unclear description of the concrete method of RNA-seq in the manuscript. The complete method is described below. The raw reads was cleaned by Fastp and evaluate by FastQC program [2]. The hisat2 was used to mapping reads to reference genome from SilkDB3.0 [3-6]. The Stringtie was used to assemble the novel transcript based on the mapping result [7]. Finally, the featureCounts was used to get raw count of every gene. All the program was used with default parameter. FPKM was calculate based on the formula and calculated in R:

This section has been revised (page 3: lines 142-146, page 4:165-167).

  1. Chen, S., et al., fastp: an ultra-fast all-in-one FASTQ preprocessor. Bioinformatics, 2018. 34(17): p. i884-i890.
  2. International Silkworm Genome, C., The genome of a lepidopteran model insect, the silkworm Bombyx mori. Insect Biochem Mol Biol, 2008. 38(12): p. 1036-45.
  3. Mita, K., et al., The genome sequence of silkworm, Bombyx mori. DNA Res, 2004. 11(1): p. 27-35.
  4. Xia, Q., et al., A draft sequence for the genome of the domesticated silkworm (Bombyx mori). Science, 2004. 306(5703): p. 1937-40.
  5. Kim, D., et al., Graph-based genome alignment and genotyping with HISAT2 and HISAT-genotype. Nat Biotechnol, 2019. 37(8): p. 907-915.
  6. Pertea, M., et al., Transcript-level expression analysis of RNA-seq experiments with HISAT, StringTie and Ballgown. Nat Protoc, 2016. 11(9): p. 1650-67.

  1. Line 191: “sequence without the mutant” to be “wildtype” or “has the construct without mutations”.

Authors' response: The “sequence without the mutant” has been revised to be “has the construct without mutations” (line 205, page 5).

  1. Line 226, please clarify what is meant by “previous studies”, does it mean the earlier experiments in this study? Re-wording is required.

 Authors' response: Thank you for pointing this out. Previous study had been accepted by Archives of Insect Biochemistry and Physiology, but not yet published. The previous study has been added as a reference (line 243, page 6).

  1. Re-wording of “screened” it could be “analyzed” in lines 267, 269. Also “was generated” could be “resulted into …DEGs”. Line 271: “ones” to be “DEGs”. Line 268: “P adjusted” to be “adjusted p value”. Line 267: to be “transcriptomic data”.

 Authors' response: These sections have been revised (lines 268-291, page 8).

  1. Line 276: “were totally different” is too general, re-wording is required e.g. “exhibited opposite expression patterns of up- and down-regulation”.

Authors' response: This section has been revised (line 296, page 10).

  1. Table 4, the DEGs represent how much of the total transcripts detected? It is best to be mentioned. Also, was a protein family analysis done? As it could be more accurate to determine the protein function and could provide more information to the DEGs’ functions.

Authors' response: Thank you for your valuable comments. We are sorry for the unclear descriptions of the results in the previous version of the manuscript. We have added the ratio of the total transcripts in Table 4. All DEGs were listed in the Supplementary Table S1-2. The KOG annotation has been added in the manuscript (Figure 5), and the analysis of KOG has been added (page 4: lines 154-156, page 9: 303-307).

  1. Line 295, it is best to mention which metabolic pathways, and mention the number of the DEGs, rather than “some DEGs”, to be more accurate.

 Authors' response: This section has been revised (lines 300-302, page 9).

  1. Clarify why BmTex261 was selected for knockdown.

 Authors' response: Thank you for your valuable comments. BmTex261 was identified and predicted to be involved in response to AcMNPV infection by the KEGG enrichment analysis that showed it was associated with apoptosis. Moreover, among the identified apoptosis genes, there was no report about the relationship of Tex261 and virus, it is valuable to detect the role of BmTex261 in AcMNPV infection. The description has been added (line 457-460, page 16).

  1. Line 338 requires re-wording, “screened” seems to be not the most accurate word.

 Authors' response: This section has been revised in the resubmitted version (line 366, page 14).

  1. Line 370: re-wording of inhibited” to “knocked down”.

Authors' response: Thank you for your valuable comments. This section has been revised in the resubmitted version (page 15: lines 381, 385, page 17: 485).

  1. Gel A in figure 7 seems to come from 2 different gels, so better separate them, or please clarify. Also each length (2771 and 510 bp), what does each represent? Mention in the legend.

 Authors' response: Thank you for your valuable comments. Description of related strips has been added (line 374, page 14).

  1. Line 365 “different concentrations”, is it increasing concentrations”, or mention the concentrations exactly, as it is a somewhat vague statement.

Authors' response: Thank you for your valuable comments. The concentrations of CaCl2 were 0 mM, 5 mM,10 mM, and 15 mM, respectively, which are mentioned in line 392 (page 15) and indicated in Fig 9.

  1. “Different times” to be corrected throughout the manuscript e,g. line 371, to e.g. “durations”, or another accurate word. And clarification as to what this time refers to, e.g. 24 hrs after transfection?!

 Authors' response: The words have been revised in this manuscript in lines 234, 251, 372, 377, 385, 400-401. And we are very sorry that we did not explain clearly in the manuscript. The “24 h” means 24 h after virus infection, and this part has been revised in the manuscript (lines 386-389, page 14-15).

  1. Line 402, please refer to the figure it is describing.

 Authors' response: Thank you for pointing this out. The ‘Figure 3’ which referenced has been gained in the resubmitted version (line 433, page 16).

  1. Line 427, “blood”, do you mean “hemolymph”, consistent terminology is best to be used.

Authors' response: These words have been revised.

  1. Lines 447, 448, 449 need re-writing as they are too general and “screened out” is not clear.

 Authors' response: Thank you for your valuable comments. These sections have been revised in the resubmitted version (lines 482-485, page 17).

  1. Finally, include a statement at the end of the discussion to state the main conclusions of this study and highlight the importance of them and what are the future prospects of it.

 Authors' response: Thank you for your valuable comments. This section has been added (lines 488-495, page 17).

  1. Grammar and typos require proofreading throughout the whole manuscript. Abstract line 6: “transcriptome” to be “transcriptomics”, “differently” to be “differentially” here and throughout the text -also in line 276-, line 12: “screened” to be “analyzed” or a similar word, as screened is not the accurate word, line 13: “virus infection” to “viral infection” throughout the text. “Series loses” to be “serious losses” throughout the text.

Authors' response: These sections have been revised.

  1. References missing in the following lines and need to be added: Page 1, lines 35 and line 39 in the intro. Lines 65, 67. Lines 88, 93, 96. Line 227. Line 362.

 Authors' response: These sections have gained references (page 1: lines 34-36, page 2: lines 66-70, 93, page 3: lines 98, 101, page 6: line 243, page 15: line 390).

  1. Page 1 line 40: “cross-infect”, seems to be an incomplete sentence, change to be “cross-infect other host domains”.

 Authors' response: This section has been revised (line 40, page 1).

  1. Typo in line 46: to be “The BmNPV”.

Authors' response: This section has been revised (line 46, page 2).

  1. Table 1: title needs more description, as to which experiment these primers belong to, and clarification which is the 5’ and which is the 3’ end of the sequences. Best to add also which region is each amplifying. The last primer has small letters which is inconsistent with the rest of the table.

 Authors' response: Thank you for your valuable comments. We realize that in this table we don't explain the primers very well. This section has been revised in the resubmitted version. The last primer has been revised to consistent with the rest of the table.

  1. Typo in line 195 to be “functional domain”.

 Authors' response: This section has been revised (line 210, page 5).

  1. Typo in line 295 to be “KEGG”.

Authors' response: This section has been revised (line 324, page 11).

  1. Table 2 title has 2 lines, line 2 seems to be requiring to be deleted. Please revise.

Authors' response: This section has been revised (line 225, page 5).

Again, thank you very much for your valuable comments and suggestions.

Reviewer 2 Report

The manuscript entitled “Comparative transcriptome analysis of Bombyx mori (Lepidoptera) larval hemolymph in response to Autographa californica nucleopolyhedrovirus infection in differentially resistant strains” examines gene expression of silkworms infected with nucleopolyhedrovirus. The C108 strain is resistant to the virus and the p50 strain is sensitive. The gene expression patterns between the silkworm strains infected with the virus were different. The results suggest that the gene expression pattern of the infection resistant strain contribute to the resistant phenotype.

The study includes beneficial information to understand the infection resistant mechanisms of silkworm. However, the descriptions of the manuscript are problems.

Comments:

1). The authors carefully explain that the C108 strain is resistant to the nucleopolyhedrovirus and the p50 strain is sensitive in Introduction section. The reviewer feel that these two strains are infection resistant strains, but the resistant mechanisms are different.

2). The reference should be cited in page 2, lines 65-67, page 2, lines 75-77, and page 6, lines 226-227.

3). In Result section, the explanation of Fig. 1B, C, and D was nothing. The authors should be added that. The authors described “In previous studies, we found that the recombinant AcMNPV was able to infect the p50 strain by injection, but not the C108 strain.”. What is a new point of Fig.1 compared with the previous study?

4). In Result section, the authors described “Principal component analysis (PCA) analysis showed that one of the sequencing data of p50- was significantly different from the other two, and subsequent analyses excluded this group.” (page 7, lines 249-251). However, the data were not shown. The data should be shown.

5). In Result section, the authors described “Linear regression was performed for the correlation between RT-qPCR and RNA-seq data (Fig. 2A).” (page 7, lines 257-258). However, the Fig. 2A is not linear regression. Moreover, the explanation of Fig. 2C-K was nothing.

The manuscript should be changed to match the data sets.

Author Response

Authors' response: Thank you for your valuable comments. Our point-by-point responses to your comments are presented below.

1). The authors carefully explain that the C108 strain is resistant to the nucleopolyhedrovirus and the p50 strain is sensitive in Introduction section. The reviewer feel that these two strains are infection resistant strains, but the resistant mechanisms are different.

Authors' response: Thank you for your valuable comments. We are sorry for the unproper description of the resistant level of the two strains. We found that recombinant AcMNPV-eGFP could be detected in p50 by puncture, but not in C108. Therefore, the resistant level p50 should be relative to that of C108, and the description has been revised in the manuscript in line 242-243 (page 6).

2). The reference should be cited in page 2, lines 65-67, page 2, lines 75-77, and page 6, lines 226-227.

Authors' response: These references have been added in lines 66-68 (page 2), 81-82 (page 2), 245-246 (page 6).

3). In Result section, the explanation of Fig. 1B, C, and D was nothing. The authors should be added that. The authors described “In previous studies, we found that the recombinant AcMNPV was able to infect the p50 strain by injection, but not the C108 strain.”. What is a new point of Fig.1 compared with the previous study?

Authors' response: Thank you for your valuable comments. The other three groups were consistent with normal silkworm (Fig. 1B-D). This section has been revised in lines 247-248 (page 6). In previous studies, silkworms infected recombinant AcMNPV by puncture was detected at the molecular level in our laboratory, but not reported at the individual level. In this study, the symptoms of AcMNPV infection in silkworm were analyzed at the individual level to distinguish them from those of BmNPV.

4). In Result section, the authors described “Principal component analysis (PCA) analysis showed that one of the sequencing data of p50- was significantly different from the other two, and subsequent analyses excluded this group.” (page 7, lines 249-251). However, the data were not shown. The data should be shown.

Authors' response: Thank you for your valuable comments. We realized that some of the descriptions of the results were not clear in the previous version of the manuscript. We have added the PCA data in Supplementary Figure S1(lines 266-268, page 7). At the same time, the deleted data has been uploaded to the SRA database (SRR15247048).

5). In Result section, the authors described “Linear regression was performed for the correlation between RT-qPCR and RNA-seq data (Fig. 2A).” (page 7, lines 257-258). However, the Fig. 2A is not linear regression. Moreover, the explanation of Fig. 2C-K was nothing.

 Authors' response: Thanks to your reminding, we realized that the drawing referenced in the manuscript was incorrectly marked, and this part has been modified (lines 274-279, page 7).

Again, thank you very much for your valuable comments and suggestions.

Reviewer 3 Report

In this paper, the authors performed transcriptomic analysis.

However, they did not show sufficient information to ensure reproducibility. First, they performed RNAseq, but they did not show accession IDs of low sequence data, which are given from SRA in NCBI. They also omit method of RNAseq data analysis (How to assemble? Trinity? Hisat2 ant stringtie? and version of these software must be added. How to calculate FPKM? RSEM?). The authors must provide data of ID list with sequences. Thus I can not assess this manuscript.

Several critical data are not shown.

Lines 248-252 They performed PCA analysis, but they did not show the results. Why ?

In table 4, they showed DEGs, however, all DEG list did not show. They must show the list in Supplemental materials. Also, annotation results by Blast2go must be shown.  

 In table 5, they showed GeneID. What GeneID? author must show these ID with sequence data, as I described above. 

Wholly , this manuscript is too immature to publish.

Author Response

Authors' response: Thank you for your valuable comments. Our point-by-point responses to your comments are presented below.

However, they did not show sufficient information to ensure reproducibility. First, they performed RNAseq, but they did not show accession IDs of low sequence data, which are given from SRA in NCBI. They also omit method of RNAseq data analysis (How to assemble? Trinity? Hisat2 ant stringtie? and version of these software must be added. How to calculate FPKM? RSEM?). The authors must provide data of ID list with sequences. Thus I can not assess this manuscript.

Authors' response: Thank you for your valuable comments. The raw data has already been uploaded to the SRA database (SRR15247045-SRR15247052), and the reviewer link was as here: https://dataview.ncbi.nlm.nih.gov/object/PRJNA749712?reviewer=e9ifu3264jbhqgnpu16qfn3jhh.

This section has been added in lines 503-506 (page 17).

We are very sorry for the unclear description of the concrete method of RNA-seq in the manuscript. The complete method is described below, The raw reads was cleaned by Fastp and evaluate by FastQC program [1]. The hisat2 was used to mapping reads to reference genome from SilkDB3.0 [2-5]. The Stringtie was used to assemble the novel transcript based on the mapping result [6]. Finally, the featureCounts was used to get raw count of every gene. All the program was used with default parameter. FPKM was calculate based on the formula and calculated in R:

This section was gained in lines 142-146 (page 3), 165-67 (page 4).

  1. Chen, S., et al., fastp: an ultra-fast all-in-one FASTQ preprocessor. Bioinformatics, 2018. 34(17): p. i884-i890.
  2. International Silkworm Genome, C., The genome of a lepidopteran model insect, the silkworm Bombyx mori. Insect Biochem Mol Biol, 2008. 38(12): p. 1036-45.
  3. Mita, K., et al., The genome sequence of silkworm, Bombyx mori. DNA Res, 2004. 11(1): p. 27-35.
  4. Xia, Q., et al., A draft sequence for the genome of the domesticated silkworm (Bombyx mori). Science, 2004. 306(5703): p. 1937-40.
  5. Kim, D., et al., Graph-based genome alignment and genotyping with HISAT2 and HISAT-genotype. Nat Biotechnol, 2019. 37(8): p. 907-915.
  6. Pertea, M., et al., Transcript-level expression analysis of RNA-seq experiments with HISAT, StringTie and Ballgown. Nat Protoc, 2016. 11(9): p. 1650-67.

  • Several critical data are not shown.

Lines 248-252 They performed PCA analysis, but they did not show the results. Why ?

Authors' response: Thank you for your valuable comments. We realized that some of the descriptions of the results were not clear in the previous version of the manuscript. We have added the PCA data in Supplementary Figure S1 in page 7, lines 266-268.

In table 4, they showed DEGs, however, all DEG list did not show. They must show the list in Supplemental materials. Also, annotation results by Blast2go must be shown.  

Authors' response: Thank you for your valuable comments. We realized that the DEGs list is necessary to our manuscript. We have added all DEGs list in Supplementary Table S1-2.

 In table 5, they showed GeneID. What GeneID? author must show these ID with sequence data, as I described above. 

Authors' response: Thank you for your valuable comments. We are very sorry that it is our fault that we did not explain this problem in the manuscript. The GeneID is corresponding to SilkDB3.0 database [2-4]. This explanation has been added to the method (line 143, page 3).

  1. International Silkworm Genome, C., The genome of a lepidopteran model insect, the silkworm Bombyx mori. Insect Biochem Mol Biol, 2008. 38(12): p. 1036-45.
  2. Mita, K., et al., The genome sequence of silkworm, Bombyx mori. DNA Res, 2004. 11(1): p. 27-35.
  3. Xia, Q., et al., A draft sequence for the genome of the domesticated silkworm (Bombyx mori). Science, 2004. 306(5703): p. 1937-40.

Again, thank you very much for your valuable comments and suggestions.

Round 2

Reviewer 1 Report

The authors have addressed the earlier comments and modified the manuscript.

Reviewer 2 Report

Thank you very much for your response.